# A Comprehensive Review of Conventional and Deep Learning Approaches for Ground-Penetrating Radar Detection of Raw Data

Xu Bai [1,*], Yu Yang [1], Shouming Wei [1], Guanyi Chen [1], Hongrui Li [1], Yuhao Li [1], Haoxiang Tian [1], Tianxiang Zhang [1] and Haitao Cui [2]

1    School of Electronics and Information Engineering, Harbin Institute of Technology, Harbin 150006, China; hit_seie_yy@163.com (Y.Y.); weishouming@hit.edu.cn (S.W.); chenguanyi@gmail.com (G.C.); lsht33@126.com (H.L.); hitliyh@foxmail.com (Y.L.); t593695287@163.com (H.T.); z3dian14@163.com (T.Z.)

2    Dalian Zoroy Technology Development Co., Ltd., Dalian 116085, China; haitao_cui@sina.cn

\*    Correspondence: x_bai@hit.edu.cn

**Abstract:** Ground-penetrating radar (GPR) is a nondestructive testing technology that is widely applied in infrastructure maintenance, archaeological research, military operations, and other geological studies. A crucial step in GPR data processing is the detection and classification of underground structures and buried objects, including reinforcement bars, landmines, pipelines, bedrock, and underground cavities. With the development of machine learning algorithms, traditional methods such as SVM, K-NN, ANN, and HMM, as well as deep learning algorithms, have gradually been incorporated into A-scan, B-scan, and C-scan GPR image processing. This paper provides a summary of the typical machine learning and deep learning algorithms employed in the field of GPR and categorizes them based on the feature extraction method or classifier used. Additionally, this work discusses the sources and forms of data utilized in these studies. Finally, potential future development directions are presented.

**Keywords:** ground-penetrating radar; detection; classification; machine learning; deep learning; raw data

## 1. Introduction

The application of radar principle to earth exploration was proposed earlier, as early as 1910, by Leim bach G. et al. in the form of patents. However, since the structure of underground media is much more complex than that of air, early ground-penetrating radar (GPR) was limited to applications in media such as ice and rock salt mines with weak electromagnetic wave absorption ability. With the improvement of the signal-to-noise ratio (SNR) of the instrument and the development of modern data processing methods, the application of GPR technology began to expand gradually after the 1970s. Since then, ground-penetrating radar has been applied to the study of lunar surface structure (L.J. Porcello, 1974) [1] to detect the thickness of weak loss media and interlayers in rock and mineral layers (Unterberger R.R., 1992) [2]. In the late 1970s, the application of GPR was gradually extended from weak lossy media such as ice and salt to lossy media such as soil, coal, and rock. In addition to the GPR used on the ground, there is another type of borehole GPR. It places the transmit and receive antennas in the borehole and it can build an electromagnetic model based on the wave interaction to perform tomography of the underground structure. Different from the GPR used on the ground, different shapes of targets in the radar map of this radar will present different characteristics. Image processing methods for borehole GPR are not covered in this paper. At present, GPR, as a nondestructive evaluation technique, has been widely applied to detect and classify buried objects for infrastructure maintenance, archeological surveys, military application, and

other geoscience studies. In many application fields of GPR, a significant step of data processing is the detection and classification of buried objects and underground structures such as rebar, landmines, pipelines, bedrock, and underground voids. Although GPR has advantages in a relatively long distance of detection for the strong penetrating power of electromagnetic, it imposes an important drawback: the images produced by GPR are not as visually intuitive and high-resolution as some other techniques so that conventional manual operation requires skilled technicians and high cost of time.

To reduce the difficulty of classification, some traditional image processing algorithms are utilized in the GPR data processing, such as background removal, filtering, and gain control. Except for those preprocessing methods, Hough transform (HT) has been widely used in hyperbola classification [3]. Then, as machine learning (ML) achieved promising results in many other fields, ML methods, including support vector machine (SVM), neural network (NN), hidden Markov model (HMM), K-nearest neighbors (K-NN), and dictionary learning (DL), have become a hotspot of GPR study. In the recent five years, convolutional neural network (CNN) and other deep learning algorithms have also been tentatively applied in the data processing of GPR system.

Though there are still some problems in the practical application, it is commonly believed that those ML methods, instead of manual recognition, will be a trend of development in the recognition and classification of GPR targets. Although these methods have achieved good results in the detection and classification of underground structures, there are still several difficult problems to complete the automatic recognition of underground targets:

- The selection of parameters in the process of image preprocessing still depends on manual work. Due to the complex structure of underground medium, there is a lot of clutter in the echo image of GPR. The plot scale parameter needs to be adjusted to enhance the target to make the target easier to detect. However, the optimal setting of this parameter is influenced by many factors, such as soil type and water content, and is very dependent on experience. It is difficult to set up automatically by computer.
- There is a lack of standard datasets for GPR images. There is currently no standard dataset in the GPR field. The existing network models are trained by simulation data or GPR data collected by researchers themselves. The trained model may not be applicable to all soil conditions.
- The amount of computation and computational complexity are high. Due to the large amount of clutter in GPR images, the computation is large and complex in the training process. How to reduce the complexity of the model while ensuring the detection performance is also a problem to be solved.

Travassos et al. reviewed the artificial neural networks (ANN) and some other ML techniques applied to GPR before 2018 [4], and they listed the techniques from locating and testing to imaging and diagnosis. Another work discussed advances in deep learning applications in ground-penetrating radar, which also contained many state-of-the-art deep learning algorithms utilized in the domain of image processing [5]. However, recognition and classification is an important and relatively independent step in image processing. In addition, as the GPR techniques as well as the ML algorithms develop, many new methods are being introduced into the detection of a target. It is necessary to make a review of all these methods.

This work focuses on the ML and deep learning methods used in the detection and classification step of GPR data interpretation. The principle of GPR imaging will be introduced in Section 2. Section 3 will discuss the conventional ML algorithms used in the detection and recognition of GPR targets. In Section 4, CNN and other deep learning methods will be presented.

## 2. The Principle of GPR

GPR works by actively transmitting an electromagnetic pulse to an underground target, which is reflected when the electromagnetic wave encounters a change in the

electromagnetic characteristics of the underground. The GPR system then receives the reflected wave. These waves will be converted into digital signals carrying information about underground targets that can be processed by the host computer. The signals are shown as one-dimensional (A-scan), two-dimensional (B-scan), or three-dimensional (C-scan) data. The three scanning modes are shown in Figure 1.

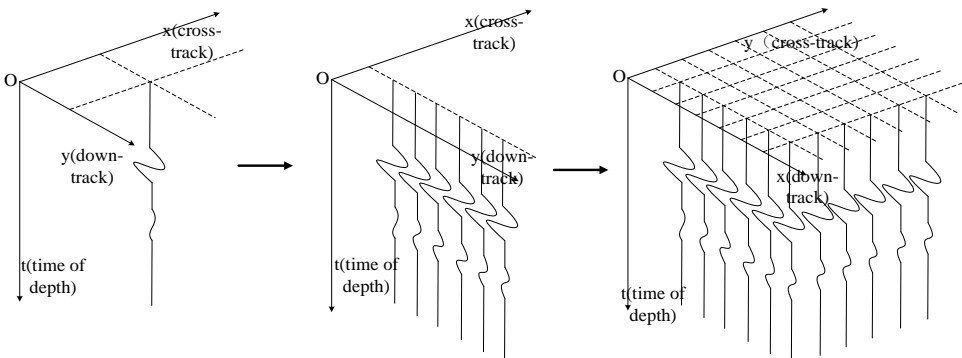

**Figure 1.** A-scan, B-scan, and C-scan data.

A-scan is the echo information of GPR system obtained by single sampling at a certain space point. It can provide accurate electromagnetic characteristics of underground objects at the current location.

The B-scan is composed of multiple equally spaced A-scan data along the direction of the scan path. It can obtain information on the path. If there is a target underground in a certain area on the path, it will cause a sudden change in the permittivity, resulting in a hyperbolic feature image [6]. Parabolic features are often used to detect subsurface objects.

The C-scan image is created by combining multiple parallel B-scan images. Three-dimensional information of underground objects can be obtained from C-scan.

By processing and analyzing these data, the underground objects can be detected and classified.

When detecting underground targets, taking B-scan data as an example, consider an ideal point target $A(x_0, y_0)$ underground. This is shown in Figures 2 and 3. Suppose that when the GPR system moves along the positive direction of the $x$-axis, its distance from target $A$ is $d$, then:

$$d = \sqrt{y_0^2 + |x - x_0|^2} \tag{1}$$

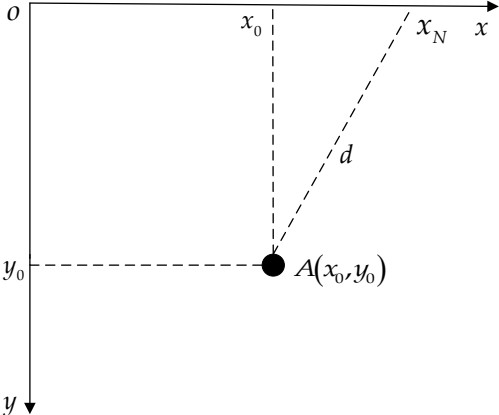

**Figure 2.** Schematic of a point-like target.

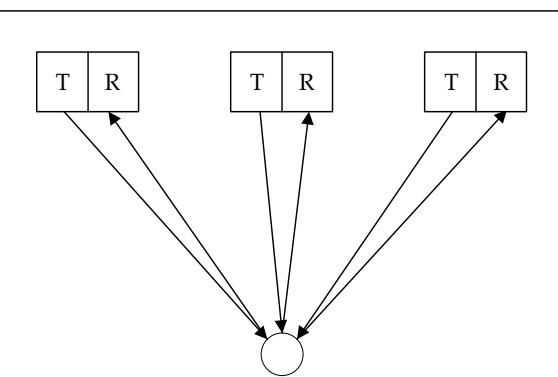

**Figure 3.** Point target detection.

For this case, the transmitting and receiving delay t of the electromagnetic wave is as follows:

$$t = \frac{2d}{v} = \frac{2\sqrt{y_0^2 + |x - x_0|^2}}{v} \tag{2}$$

Rearranging this formula yields:

$$\frac{t^2 v^2}{4} - |x - x_0|^2 = y_0^2 \tag{3}$$

Substituting $y_0 = vt_0/2$ into the formula yields:

$$\frac{t^2}{t_0^2} - \frac{|x - x_0|^2}{\frac{t_0^2 v^2}{4}} = 1 \tag{4}$$

The dielectric constant of the subsurface medium is assumed to be $\varepsilon_r$, and the magnetic properties of the underground medium are ignored. Then, the wave velocity of the electromagnetic wave in the underground medium is $v = c/\sqrt{\varepsilon_r}$. Plugging it into Equation (4) yields:

$$\frac{t^2}{t_0^2} - \frac{|x - x_0|^2}{\frac{t_0^2}{4}\left(\frac{c}{\sqrt{\varepsilon_r}}\right)^2} = 1 \tag{5}$$

Observing the expression, it can be known that the point target has hyperbolic characteristics in the B-scan image. This is shown in Figure 4.

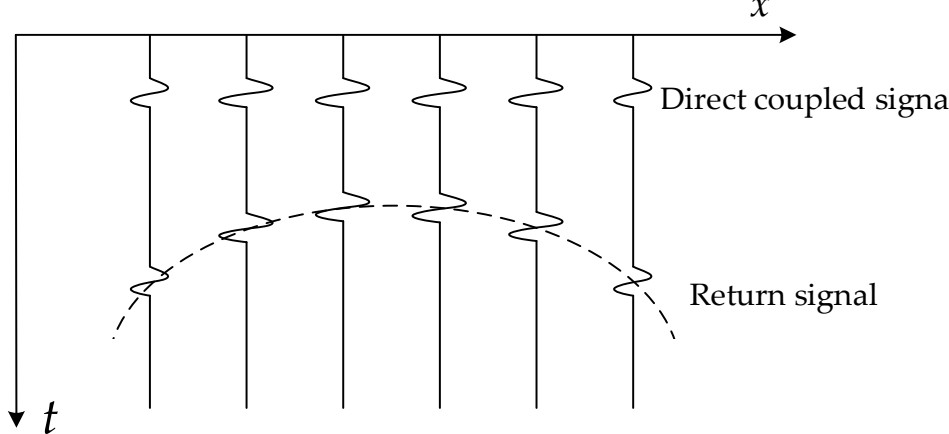

**Figure 4.** Echo image of point target.

When the target is extended to a spherical target with radius *R*, it is easy to obtain its curve equation as follows:

$$\frac{\left(t + \frac{2R}{v}\right)^2}{\left(t_0 + \frac{2R}{v}\right)^2} - \frac{(x - x_0)^2}{\left(\frac{t_0 v}{2} + R\right)^2} = 1 \tag{6}$$

When the GPR system moves at a constant speed along the survey line, the target will show a hyperbolic feature in the data form of B-scan. When performing object recognition and detection, the part of the image with hyperbolic features is also detected. Similarly, the targets under C-scan exhibit hyperboloid features.

## 3. The Application of Conventional ML Algorithms

Although the basic concepts of many conventional ML methods were proposed many years ago, those methods are still widely used because of their advantages, such as simple structure and low computation complexity. Most of the classification methods based on conventional ML consist of two steps: feature extraction and classifier. Some classic ML algorithms such as SVM and K-NN are classifiers, while others such as DL are applied in the feature extraction.

### 3.1. Support Vector Machine (SVM)

The support vector machine (SVM) is a machine learning method used for classification and regression analysis, which was originally proposed by Vapnik et al. in the 1990s. The SVM method maps the data points in the sample space to a high-dimensional space, where an optimal hyperplane is identified to maximally separate the data points of different classes. The SVM classifier is shown in Figure 5. This hyperplane is known as the separating hyperplane. The "support vectors" in SVM are the data points that are closest to the separating hyperplane, which determine the position and orientation of the hyperplane. The primary objective of SVM is to maximize the distance between the support vectors and the separating hyperplane, i.e., the classification margin, while ensuring the accuracy of the classification. This distance is called the margin, and SVM aims to identify the hyperplane with the largest margin. When the data are not linearly separable, SVM can perform nonlinear classification in a high-dimensional space by utilizing a kernel function to map the data points to a higher-dimensional space.

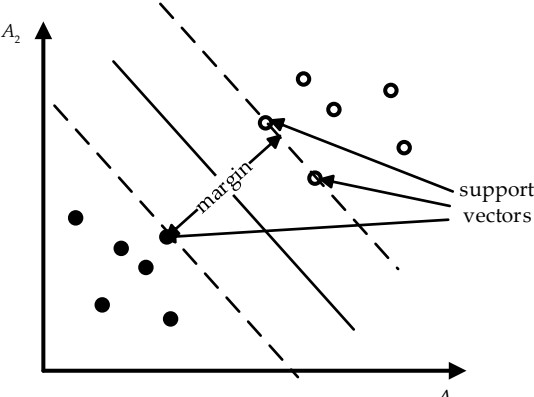

**Figure 5.** SVM classification model.

SVM is one of the most popular classifiers due to its excellent generalization capability. In 2008, Pasolli et al. applied generic algorithm (GA) in the detection of the underground target, then proposed a method consisting of a feature extraction strategy aimed at the dielectric characteristics of the target and a classifier based on SVM to recognize the material of buried objects [7]. In 2013, El-Mahallawy et al. also used the discrete cosine transform

(DCT) features of GPR data and SVM for material classification and compared it to other conventional methods [8]. In the same year, Xie, X.Y. et al. used SVM algorithm to identify concrete internal voids and studied the accuracy of four kernel functions [9]. In 2018, a study used the H-Alpha decomposition (a polarimetric decomposition) to extract the feature and trained an SVM classifier [10]. Tbarki et al. developed a COSVM classifier and, different from these mentioned methods based on B-scan data, they took some points of A-scan data as a feature vector for detection of a landmine [11]. In addition, it is proven in this work that COSVM kept a balance between the classification accuracy and the cost of training time, compared with one-class SVM (OSVM), Mahalanobis one-class SVM (MOSVM), and two-class SVM. In the same year, Genc, A. et al. proposed a method to distinguish targets by energy and perform mine identification [12].

Since SVM is widely used in the detection and classification of GPR targets and regarded as a highly adaptable method, it plays an important role in the comparative research of various feature extraction strategies. A study compared the performance of feature extraction based on histogram of oriented gradients (HOG) and n-Row average subtraction (nRAS) with SVM and K-NN classifiers [13]. In addition, in the tests with SVM, 3Ras performed better in the positive object discrimination ratio (POD) than nRAS and the original image cropped without any normalization. Presented in [14] is another study made by the same group as [13]; they proposed a novel fast feature extraction method that converted the interest region into a feature vector (FV) and combined it with nRAS and background removal technique (BRT). They compared these two fusions of methods, which were named as nR-FV and BRT-FV, respectively, with FV, nRAS, and HOG. When implemented with SVM, 3RAS remained the best performance, while nR-FV had less feature generation time. In 2017, Sakaguchi et al. made a comparison of feature representation methods shown in Table 1 for automatic buried threat detection (BTD) [15]. In this work, feature extraction methods, which had been applied to BTD with GPR or were potentially promising, were divided into three categories: gradient feature extraction methods, binary comparison methods, and pixel representations. The researchers chose partial least squares discriminant analysis (PLSDA), linear SVM, and nonlinear SVM as classifiers and nonlinear SVM achieved the best performance across all these extraction methods. Except for the overall performance, this work calculated the correlation between the confidence produced by each feature method and discussed the fusion of confidences from the classifier and prescreening. Another work in 2019 discussed the performance of two kinds of edge histogram descriptors (EHD), Log Gabor (an improvement of Gabor filter), gprHOG (an HOG-based methods for GPR), and spatial edge descriptors (SED) in BTD [16]. The other feature extraction methods used SVM classifier, while gprHOG was implemented with two random forest classifiers. The curves of receiver operating characteristics (ROC curve, a curve jointly specifies the rate of detection and level of false alarm) when these algorithms were utilized in the detection of threats buried deep and not deep were presented. Log Gabor had the best performance on deeply buried threats substantially, and SED outperformed the other algorithms over most of the curves on threats buried not so deep. In the analysis part, the researcher discussed the similarity of the algorithm design practices and the difference in the conception of these methods.

*3.2. K-Nearest Neighbors (K-NN)*

K-nearest neighbor (K-NN) algorithm is a commonly used supervised learning method in machine learning, which is a nonparametric lazy learning method. The working principle is that, for a given test sample, the K-nearest training samples are found according to some distance metric, and the prediction is made based on the information of these training samples. The principle of K-NN algorithm is shown in Figure 6.

To implement the K-NN algorithm, we first need to determine two parameters, the algorithm hyperparameter K and the distance metric of the model vector space.

K is the only hyperparameter in the K-NN algorithm. The selection of the value of K has a great impact on the performance of the algorithm. When the K value is small, only

the training samples that are similar to the input will play a role in the prediction result. At this time, the approximation error of the algorithm is relatively small. However, because the prediction results are sensitive to the neighboring points, when the neighboring points are noise points, the prediction will have a large deviation, resulting in an over-fitting phenomenon. Similarly, when the value of K is selected to be large, points that are far away will also have an impact on the prediction results. At this time, the algorithm has good robustness, but it is prone to under-fitting.

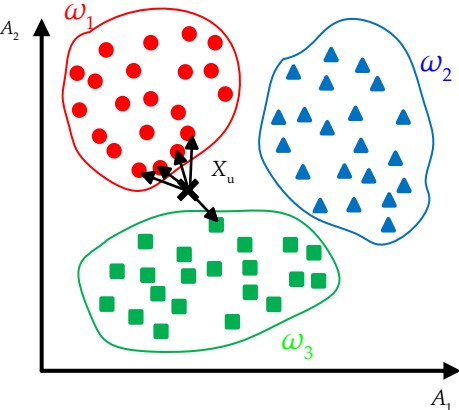

**Figure 6.** K-NN classification model.

The distance measure indicates how similar two points in the sample space are. Commonly used distance metrics are Manhattan distance, Euclidean distance, Chebyshev distance, Minkowski distance, cosine distance, etc.

K-NN is widely used in the classification and recognition of GPR images. As early as 2000, K-NN algorithm has been applied to GPR image detection [17]. In 2007, Frigui et al. proposed a method consisting of EHD and a possibilistic K-NN classifier for landmine detection with GPR, as a possibilistic K-NN classifier can discriminate between neighbors with equal distance while standard K-NN, weighted K-NN, and fuzzy K-NN cannot [18]. In addition, it was presented in this work that possibilistic K-NN also performed better than crisp K-NN and fuzzy K-NN over ROC curves for the detection of landmines. In [13,14], K-NN had better results than SVM overall, which was probably caused by the decision pattern by two classifiers, as K-NN decided according to the votes from nearest neighbors while SVM decided with consideration of the entire dataset. In 2019, Elsaadouny, M. et al. used moving average background subtraction (MA-BS), DC-offset removal, and the subtraction and weight (SaW) methods for data augmentation and K-NN for classification [19]. Another study in 2019, using features of row and column vectors to produce a twin gray statistics sequence (TGSS) as an input of classifiers, applied K-NN together with rotation forest, logistic regression (LR), and some other algorithms in the hyperbola classification [20].

### 3.3. Hidden Markov Model (HMM)

In contrast to a normal Markov model, in a hidden Markov model, the state is not directly visible but some variables affected by the state are. Each state has a probability distribution over the possible output symbols. So, the sequence of output symbols can reveal some information about the state sequence. Hidden Markov model is a model based on probability calculation; it is a special dynamic Bayesian network model; it is a double random process composed of general random process and Markov chain. Different from SVM and K-NN, HMM produces time sequence of random observations as a function of the state. The word hidden in HMM comes from the attribute that the actual state is hidden or not observable. Given that an observation vector is observed at a certain time, there are probabilities that the process is in each state. The actual state of the process is hidden. In other words, each state is represented by a probability density function, which can either be continuous or discrete. If time sequences are known, the hidden state can be eliminated.

In 1999, Gader et al. introduced HMM into the GPR detection for the first time; it was used to model the time-varying signatures caused by the landmines as the vehicle moved [21]. Two models are established in this paper, one for background and the other for mines. Discrete HMM was used in the experiment, the Viterbi algorithm is used to look back at the model state, the fuzzy clustering algorithm is used to find the state, and the Baum–Welch algorithm is used to estimate the model parameters. Then, in 2001, they made an expansion of the previous work in testing and evaluation of methods and used both discrete and continuous HMM [22]. The B-scan images were divided into slices, and their feature vectors were 16-dimension vectors defined by the values of the positive or negative diagonal and antidiagonal arrays in the neighborhoods of their respective maxima. The discrete HMM was trained by the Baum–Welch (BW) algorithm and the continuous HMM was initialized using clustering methods and learned using the BW algorithm. The continuous HMM performed slightly better than the discrete HMM, and the fusion of the two resulted in a better performance than the individual algorithms. In 2005, Frigui et al. made progress in that they developed a complete real-time software system for landmine detection based on continuous HMM and applied it to data acquired from outdoor test sites [23]. Two corrective training scenarios were used to improve the overall performance by about 10%. Then, [24] used Gabor filter as a feature extraction strategy instead of the feature vectors in [22], which assumed that the signatures of targets had a rising edge or a falling edge with a certain orientation. In 2011, Missaoui et al. applied the multi-stream discrete HMM (MSDHMM) to the classification of landmines [25]. The core idea of MSDHMM is the fusion of different sensors or different features and classifiers of one sensor. In this work, they made use of a fusion of Gabor and EHD and adopted classifiers based on distance and possibility, respectively. The result of this study showed that the structure of MSDHMM improved the ability of discrimination of HMM.

Ground-penetrating radar systems achieved high detection rates but at the expense of high false alarm rates. In 2014, Rebecca M. Williams et al. used a combination of SVM and HMM with down sampling, a preprocessing step that is unbiased and suitable for real-time analysis and detection [26]. The HMM is utilized as a pre-screener that evaluates all traces of the file sequentially and the SVM is subsequently used as a confirmer that evaluates only those traces selected by the HMM. The combined HMM–SVM method retains all of the correct classifications by the SVM and reduces the false positive rate to 0.0007. Receiver operating characteristic (ROC) curve is a co-ordinate diagram composed of false positive rate as the horizontal axis and hit probability as the vertical axis. ROC curve can reflect the discrimination ability of the model. In 2015, Anis Hamdi et al. proposed a novel ensemble HMM classification method that is based on clustering sequences in the log-likelihood space [27]. The eHMM uses multiple HMM models and fuses them for final decision making. Whether compared with discrete HMM or continuous HMM, the ROC performance of eHMM is superior. Another work about the application of HMM in GPR combined HMM with multiple-instance learning (MIL) [28]. The depth bins below and above the target had different characteristics and this multiple-instance HMM (MiHMM) method could incorporate the ambiguity in the individual sample labels. It was presented in this study that MiHMM had better ROC performance than standard EM-HMM and nonparametric Bayesian multiple-instance learning (NPBMIL) on discrete features, while NPBMIL outperformed the other two methods on HOG features.

### *3.4. Artificial Neural Network (ANN)*

An artificial neural network (ANN) is a mathematical model or computational model that imitates the structure and function of a biological neural network and is used to estimate or approximate functions. In most cases, the artificial neural network can change the internal structure on the basis of external information and it is an adaptive system, that is, it has a learning function. Like other machine learning methods, neural networks have been used to solve problems that are difficult to solve by rule-based programing. ANN has

a distinguishable structure that is composed of multiple layers, and the conventional ANN usually consists of input layer, hidden layer, and output layer.

The structure of a single neuron is shown in Figure 7; it can be seen that the function of a single neuron is to obtain a scalar result through a nonlinear transfer function after obtaining the inner product of the input vector and the weight vector. Its function is to divide an n-dimensional vector space into two parts with a hyperplane (called the judgment boundary) and then, arbitrarily given an input vector, the neuron can determine which side of the hyperplane the vector is on, which actually completes the classification function.

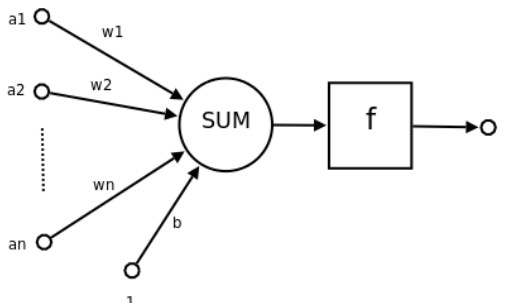

**Figure 7.** Schematic diagram of a single neuron structure.

A single-layer neural network can be obtained by combining a limited number of single neurons. The schematic diagram of its structure is shown in Figure 8; this is the most basic form of neuron network. The input vectors of all neurons are the same vector, and each input vector will also be connected to all neurons. Since each neuron produces a scalar result, the output of a single layer of neurons is a vector whose dimension is equal to the number of neurons.

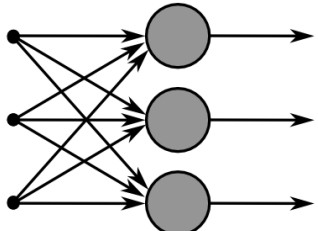

**Figure 8.** Schematic diagram of a single-layer neural network structure.

As CNN and other popular deep learning methods have structural differentiation from conventional ANN, their application in GPR classification will be discussed in Section 3.

In 2000, Al-Nuaimy extracted the spectral features of GPR scan, and then divided these feature vectors into training and testing set [29]. They proposed an NN to learn the instances in the training sets to construct a rule for the labelling of regions with or without buried objects. Another study in the same year used shape filtering to extract the hyperbolic signatures from the binarized B-scan images and developed a two-layer feedforward network classifier [30]. By combining simple data preprocessing rules and neural network image interpretation, a robust pipeline feature space detection method is proposed. Satisfactory results were obtained through experiments. Gader et al. proposed a complex recognition system of landmines and morphological shared-weight neural networks (MSNN) were utilized in this system, of which the input was down-track slices of B-scan data [31]. The reason for using this network is its ability to simultaneously learn feature extraction and classification parameters, further improving efficiency. The method based on fuzzy set-based fusion of the information sources results in significant reductions in false rates. In 2005, a multi-layer perceptron (MLP) network was applied to the location of steel reinforcement in rebar [32]. The reason for choosing MLP is that it has low complexity in terms of node count, network training, and application and the data can be preprocessed

before MLP network analysis. The main reason is that this network could learn with the simplified dataset that had been processed by edge detection. Then, in 2010, a standard multilayer neural network trained by trapezoidal image sections with the apex and one branch of the hyperbola was applied to the automatic detection of reflection hyperbolas in B-scan images [33]. The neurons are no longer fully connected to the input vector; each one rather receives just a small area of the presented image section. Adjacent neurons thereby are responsible for adjacent areas, which are freely configurable and may overlap arbitrarily. At the same time, using a half trapezoid as the shape for the training data and a two-dimensional analysis via receptive fields and local connections in a multilayer network, it is possible to reliably detect the apexes even under conditions in which nearby located objects lead to interfering or disturbed hyperbolas.

As the computation power of computers developed, the input data volumes of neural networks became larger and more complex. In 2013, Singh et al. used a neural network with two layers and 1200 input nodes to classify the hyperbolic signature of buried objects [34]. The experimental error is within an acceptable range. A study in 2014 began to use a sliding window to extract features as the input of a neural network for landmine and unexploded ordnance (UXO) detection [35]. Under high computational complexity, the neural network technique is a more robust algorithm in terms of the accuracy and the false positive and false negative error rates. Then, Szymczyk et al. proposed a novel ANN structure that used the Laplace transform instead of ordinary weights and activation function of an artificial neuron for classification of geological structure [36]. After 2015, cellular neural networks (CNN) began to emerge.

### 3.5. Dictionary Learning (DL)

A vector is sparse when most of its elements are zero. With sparse coding, the input signal is decomposed into basic elements—atoms. Sparse representation (SR) is an effective way to extract the mid-level or high-level features in images [37], which assumes that natural signals can be represented linearly by a sparse linear combination of atoms of a dictionary. It is written as $X = D\alpha$. In the formula, $X$ represents the signal, $D$ represents the dictionary, and $\alpha$ represents the sparse matrix. Sparse means that there are only a few non-zero values in $\alpha$. In sparse representation, the dictionary can be thought of as a kind of transformation domain. In this transformation, the signal representation is sparse. This problem can be mathematically modeled as follows:

$$\min_{D,X}\{\|\alpha_i\|_0\} \quad s.t. \|X - D\alpha\|_F^2 \leq \varepsilon \tag{7}$$

There are two ways to generate dictionaries. One is based on a preconstructed model, such as Fourier base, wavelet base, discrete cosine change base, etc. The dictionary generated in this way has good structure and fast numerical calculation. However, it has a strong limitation that it can only represent a certain type of signal. The other is based on the learning model through training samples to obtain the dictionary. Dictionaries obtained in this way have better adaptability. The process of solving the dictionary based on the available data is called dictionary learning (DL). In addition, SR is usually implemented with conventional classifiers such as SVM. In 2013, Shao et al. applied an SR method that employed an overcomplete Gabor dictionary to signal classification in A-scan data [37]. They compared this algorithm to wavelet decomposition and K singular value decomposition (K-SVD) and the result indicated that the proposed method could achieve a better approximation of the GPR traces than the other two methods. In 2016, Terrasse et al. built a dictionary of theoretical pipe signatures, then used the correlation between the B-scan data and each atom from the dictionary as input features of SVM classifier [38]. The proposed method in this paper outperformed HOG and Canny Edge detection algorithm in the rate of detection. In 2017, Giovanneschi et al. utilized an online dictionary learning (ODL) method with an SVM classifier in the mine detection and proved that ODL reduced learning time by 94% and increased the rate of target detection by 10% over the

K-SVD methods [39]. Another study in 2017 applied a label-consistent K-SVD (LC-KSVD) algorithm, which was specifically designed for the extraction of useful features instead of reconstruction of images to deal with the BTD problems [40]. It needed no additional classical classifier and had better performance in proportion of target detection for BTD than HOG with a linear classifier and the HOG with a nonlinear classifier (a random forest). A novel method named deep dictionary learning was introduced into the classification of embedded objects in B-scan images in 2018 [41]. The algorithm proposed in this study can shorten the data calculation time. In this multilayer deep dictionary learning structure, the atomic size of each layer of dictionary is different. In this paper, the performance of various classifiers was presented. Among all these methods, the combination of linear discriminant analysis (LDA) classifier and dictionary learning with three hidden layers had the highest classification accuracy; the detection performance reaches 94.4% without preprocessing and postprocessing, reaching more than 94%. In 2019, Drop-Off MINi-batch Online Dictionary Learning (DOMINODL), which processed the training data in mini-batches, was proposed and utilized in landmine detection. Compared with K-SVD, ODL, low-rank shared DL (LRSDL), and correlation-based weighted least square update (CBWLSU), DOMINODL was the fastest method and had performance comparable to others [42]. Recently, dictionary learning has also been applied to clutter removal in GPR images. Moreover, in 2019, Jingtao Zhao et al. proposed a GPR diffraction extraction model combining PWD method and sparse coding algorithm based on online dictionary learning to solve the problem that geological information is difficult to extract due to the shielding effect of strong reflections from underground layers [43]. Through the PWD method, a local-plane wave mathematical model is used to predict linear events, thereby removing GPR strong reflection components. The GPR diffraction with amplitude change or phase reversal and noise are filtered into the residual data, and the weak diffraction is further recovered from the residual data by the dictionary learning method, and the weak diffraction component is extracted. Compared with traditional GPR migration profiles, the algorithm provided by this study can provide more detailed information on GPR signals such as small-scale holes and edges. A 2021 study applied K-SVD dictionary learning to GPR data denoising, used the orthogonal matching pursuit (OMP) algorithm to sparsely decompose different radar signals, and trained an overcomplete dictionary with sample features. The dictionary obtained through training performs better than the fixed discrete cosine transform (DCT). The K-SVD dictionary can effectively distinguish valid data from noise in GPR data. The K-SVD dictionary learning algorithm adaptively learns the noise data and considers the intra-block information and the global observation to complete the denoising [44]. In 2020, Kumlu, D. et al. proposed a method based on robust orthogonal subspace learning (ROSL) for clutter removal. The original image is decomposed into two parts: clutter and target by ROSL. It performs better than robust principal component analysis (RPCA), go decomposition (GoDec), and other algorithms [45]. In 2022, Fok Hing Chi Tivive and others proposed the multilevel projective dictionary learning with low-rank prior (MPDL-LR) model [46], which uses a multilevel projective dictionary method to estimate radar trace signals. MPDL-LR achieves an average classification accuracy of 94.5% when using the $\varepsilon$-dragging technique, which is higher than 93.8% when using a linear classifier on the same test set. If the explosive devices are classified into high-metal and low-metal in advance, the accuracy rate of MPDL-LR predicting improvised explosive devices will be further increased to 98%. At the end of 2022, a study conducted by Siqi Li et al. based on the experiments conducted by Feng D used fast iterative shrinkage-thresholding algorithm (FISTA) [47] instead of OMP, which further improved the denoising effect and algorithm convergence speed. The processing of echo data shows that, under different noise levels, the FISTA algorithm is better than the OMP algorithm in denoising the radar signal and, with the increase in the noise, the superiority is getting bigger and bigger. In a 2023 study on the reconstruction of 2D images of buried objects from ground-penetrating radar reflection hyperbolas, Shayan Hajipour et al. found that DL methods outperform CNN methods in terms of noise measurement because sparse representations are more robust to image noise, while CNN methods work

better when dealing with fewer trajectories [48]. Because the essence of dictionary learning is to find the similarity between the dictionary atoms and the input signal and when the number of scans becomes smaller and smaller, this similarity will become more and more difficult to find. In 2023, Zhi-KangNi et al. analyzed the largest singular value, divided the atoms in the adaptive dictionary obtained by deep learning into target atoms and clutter atoms, and reconstructed the target component and clutter component in the GBR B-scan signal, respectively [49], thereby reducing the clutter interference caused by the target reflection and improving the detection accuracy. In terms of clutter suppression, the pro-posed algorithm is superior to MCA, OSTD, RNMF, and SVD in almost all scenarios and only slightly inferior to RAE and RPCA in metal pipeline scenarios. In recent years, some researchers have used the subspace decomposition method to study the stratification of underground media. Zhou, C.Y. et al. used the multiple-signal classification (MUSIC) method to generate super-resolved velocity–depth spectra for underground multilayer analysis. The proposed method outperforms traditional SAR-based methods and has great potential for use [50].

Table 1 summarizes typical cited works and classifies them into the types of feature extraction methods, classifiers, and data form.

**Table 1.** The application of conventional ML in GPR detection and classification.

| Feature Extraction | Classifier | Year | Data Form |
|---|---|---|---|
| generic algorithm (GA) | SVM | 2008, 2009, 2009 [7,51,52] | B-scan |
| discrete cosine transform (DCT) | SVM | 2013 [8] | A-scan |
| features extracted from forward simulation data | SVM | 2013 [9] | B-scan |
| HOG | SVM | 2015 [53] | B-scan |
| HOG | SVM | 2015 [54] | B-scan |
| H-Alpha decomposition | SVM | 2018 [10] | B-scan |
| some points of A-scan data | COSVM | 2018 [11] | B-scan |
| the A-scan image with the highest energy in the B-scan image | SVM | 2018 [12] | A-scan |
| HOG and nRAS | SVM and K-NN | 2018 [13] | B-scan |
| FV, nRAS, BRT and their fusion | SVM and K-NN | 2020 [14] | B-scan |
| EHD, SIFT, SURF, HOG, LBP, BRIEF, normalized pixel values, PCA and block PCA | PLSDA, linear SVM and non-linear SVM | 2017 [15] | B-scan |
| EHD, Log Gabor, gprHOG and SED | SVM and MSEK | 2019 [16] | B-scan and different sections of C-scan data |
| PCA and down sampling | K-NN | 2000 [17] | A-scan |
| EHD | K-NN | 2007 [18] | B-scan |
| MA-BS, DC-offset removal, and SaW | K-NN | 2019 [19] | B-scan |
| TGSS | K-NN | 2019 [20] | B-scan |
| certain orientation extraction | HMM | 1999, 2001, 2005 [21–23] | B-scan |
| evolutionary algorithm (EA) | HMM | 2003 [55] | B-scan |
| Gabor filter | HMM | 2007 [24] | B-scan |
| a fusion of Gabor and EHD | MSDHMM | 2011 [25] | B-scan |
| normalization, edge detection and other methods | SVM and HMM | 2014 [26] | B-scan |
| EHD | eHMM | 2015 [27] | B=scan |
| HOG | MiHMM | 2015 [28] | B-scan |

**Table 1.** *Cont.*

| Feature Extraction | Classifier | Year | Data Form |
|---|---|---|---|
| spectral features | NN | 2000 [29] | A-scan and B-scan |
| shape filtering | NN | 2000 [30] | B-scan |
| down-track slices | MSNN | 2001 [31] | B-scan |
| edge detection | MLP | 2005 [32] | B-scan |
| trapezoidal image sections | NN | 2010 [33] | B-scan |
| some points of B-scan data | NN | 2013 [34] | B-scan |
| sliding window | NN | 2014 [35] | B-scan |
| Laplace transform | LTANN | 2015 [36] | B-scan |
| SR with an overcomplete Gabor dictionary | SVM | 2013 [37] | A-scan |
| a dictionary of theoretical pipe signatures | SVM | 2016 [38] | B-scan |
| ODL | SVM | 2017 [39] | A-scan |
| LC-KSVD | SVM | 2017 [40] | B-scan |
| deep dictionary learning | no additional classifier | 2018 [41] | B-scan |
| DOMINODL | SVM | 2019 [42] | B-scan |
| ODL | no additional classifier | 2019 [43] | B-scan |
| K-SVD | no additional classifier | 2021 [44] | B-scan |
| ROSL | no additional classifier | 2020 [45] | B-scan |
| MPDL-LR | $\varepsilon$-dragging technique | 2022 [46] | B-scan |
| K-SVD | no additional classifier | 2022 [47] | B-scan |
| DL | NN | 2023 [48] | A-scan and B-scan |
| DL | no additional classifier | 2023 [49] | B-scan |
| Hough transform | Viola-Jones | 2013 [56] | B-scan |
| MUSIC | no additional classifier | 2023 [50] | A-scan |

## 4. The Application of CNN and Other Deep Learning Algorithms

Convolutional neural networks (CNN) are a type of deep learning model commonly used for image processing and computer vision tasks. The basic principles of CNN are as Figure 9.

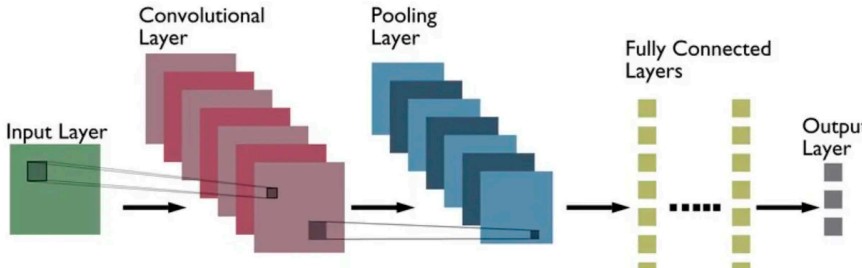

**Figure 9.** Typical CNN architecture.

- Convolutional Layers:

The convolutional layer is the core layer of a CNN. It performs convolutional operations on the input image using a set of filters (also known as kernels or convolution matrices). These filters can capture different features in the image, such as edges, textures, etc. The convolution operation involves sliding the filters over the image and calculating the dot product between the filter and the input image to generate a feature map (also known as a convolutional feature or convolutional map).

The convolutional operation between an input image (or feature map) denoted as X and a filter (or kernel) denoted as W can be mathematically represented as:

$$Convolution(X, W) = \sum (X * W) + b \tag{8}$$

where "$*$" denotes the convolution operation, "$\sum$" denotes the sum, "$b$" is the bias term, and "$X$" and "$W$" are matrices.

- Activation Functions:

Activation functions are typically applied after the convolutional layer to introduce nonlinearity. Popular activation functions used in CNN include rectified linear unit (ReLU) function, which helps the model learn more complex feature representations.

$$ReLu(x) = \max(0, x) \tag{9}$$

- Pooling Layers:

Pooling layers are used to reduce the spatial dimensions of the feature map, thus reducing computation and the number of parameters. Common pooling operations include max pooling and average pooling, which extract the most salient features from the feature map.

$$\text{Max pooling}: \text{MaxPooling}(X) = \max(X) \tag{10}$$

$$\text{Average pooling}: \text{AvgPooling}(X) = \text{average}(X) \tag{11}$$

- Fully Connected Layers:

Fully connected layers are used to transform the output of the pooling layers into the final classification or regression output. Neurons in the fully connected layer are connected to all neurons in the previous layer, and each connection has a weight that needs to be learned through training.

$$FullyConnected(X, W, b) = W * X + b \tag{12}$$

where "$X$" is the input feature vector, "$W$" is the weight matrix, and "$b$" is the bias vector.

The net input $z^{(l)}$ of layer $l$ is the activation value $a^{(l-1)}$ of layer $l - 1$, and convolution with kernel $w^{(l)} \in R^K$.

$$z^{(l)} = w^{(l)} \otimes a^{(l-1)} + b^{(l)} \tag{13}$$

The convolution kernel $w^{(l)} \in R^K$ is a learnable weight vector, and $b^{(l)} \in R$ is a learnable offset.

According to the definition of convolution, the convolutional layer has two very important properties:

Local connection: each neuron in the convolutional layer (assumed to be the $l$ layer) is only connected to the neurons in a local window in the next layer (the $l - 1$ layer), forming a local connection network. As shown in Figure 10, the number of connections between the convolutional layer and the next layer is greatly reduced, from the original $M_l \times M_{l-1}$ connections to $M_l \times K$ connections, where $K$ is the size of the convolution kernel.

Weight sharing: the convolution kernel $w^{(l)}$ as a parameter is the same for all neurons in the $l$ layer. As shown in Figure 11, the weights on all connections of the same color are the same.

Due to local connections and weight sharing, the parameters of the convolutional layer only have a $K$-dimensional weight $w^{(l)}$ and a one-dimensional bias $b^{(l)}$, a total of $K + 1$ parameter. The number of parameters has nothing to do with the number of neurons. In addition, the number of neurons in the $l$ layer is not selected arbitrarily but satisfies $M_l = M_{l-1} - K + 1$.

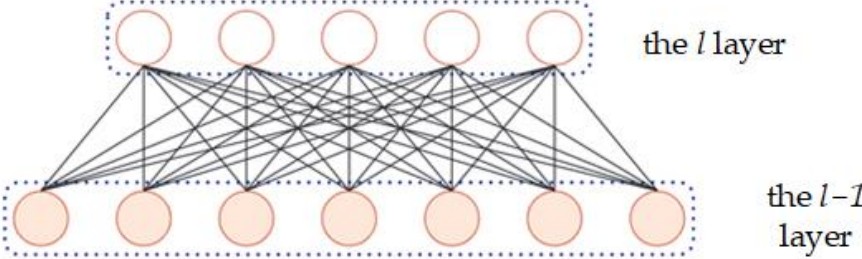

**Figure 10.** Fully connected layer.

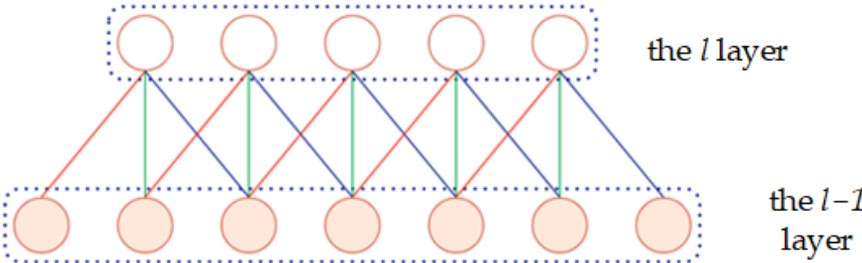

**Figure 11.** Convolutional layer.

A typical convolutional network is formed by cross-stacking convolutional layers, pooling layers, and fully connected layers. A convolutional block is consecutive *M* convolutional layers and *b* pooling layers. In a convolutional network, *N* consecutive convolutional blocks can be stacked, followed by *K* fully connected layers.

As an end-to-end algorithm, the concept of CNN has been proposed for a long time, developing from 1D structure to 2D and 3D structure. In addition, deep learning is a subset of machine learning. When the deep CNN trained by Krizhevsky and his group achieved record-breaking performance in the ILSVRC-2012 [57], it attracted great attention from researchers in different fields, and CNN has gradually become one of the most popular models of deep learning since then.

CNN was first reported to be applied in GPR classification in 2015; Besaw et al. utilized a CNN consisting of two convolutional layers and corresponding max-pooling layers in the classification of buried explosive hazards (BEH). They applied heuristics, including cross-validation, network weight regularization, and "dropout" to prevent over-training, to the CNN structure [58]. Almost at the same time, another study also presented the results of using a simplified CNN in the recognition of subsurface target [59]. This study tested different parameters of the network and provided some suggestions on parameter settings. Both of these studies showed that CNN had a promising performance in feature extraction and classification. In 2016, Besaw et al. compared the performance of CNN method and traditional algorithms including EHD and HOG for classification of BEH and false alarms [60]. It was shown in this paper that CNN outperformed the traditional methods in the probability of detection, and data augmentation could improve the performance of both methods significantly. In 2018, a nine-layer convolutional neural network developed by Sonoda, J. et al. can identify six materials with 80% accuracy in uneven underground media [61]. There are also some researchers who combine CNN with traditional machine learning methods. In 2020, U. Ozkaya et al. combined CNN with SVM to build the CSVM structure. CSVM has multiple convolutional and pooling layers and a single SVM classifier. Since the SVM layers are learned with an unsupervised approach, there is no need for backpropagation algorithm to update the weights. The proposed method improves the classification performance with low computational complexity [62]. The network structure of [62] is shown in Figure 12.

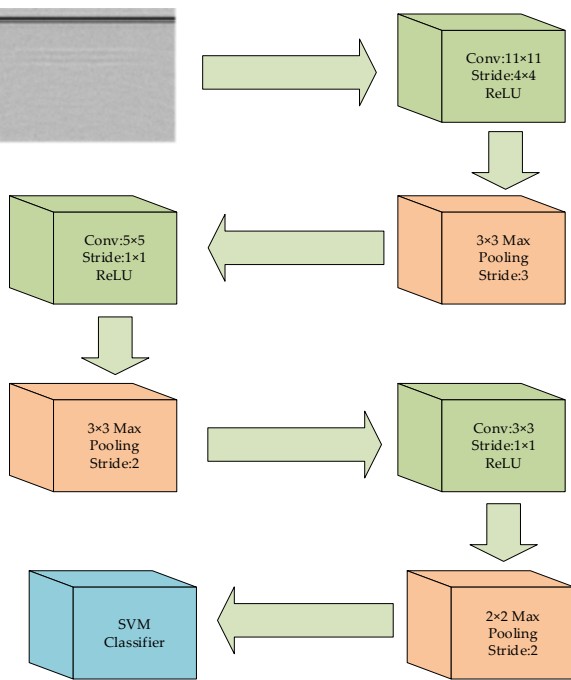

**Figure 12.** The network structure in [62].

The lack of real data for training, which can lead to an over-training problem, is always a big challenge of CNN application in GPR data. Therefore, cross-validation, weight regularization, dropout, pretraining, data augmentation, and other techniques to prevent over-training have been widely used in CNN structure for GPR. In 2017, Bralich et al. pretrained a CNN on the Cifar10 dataset and a dataset of high-resolution aerial imagery and showed that transferring weights from a pretrained CNN and fine-tuning the parameters could improve detection performance [63]. Another work in 2017 also adopted pretraining and data augmentation to CNN for BTD problems in GPR data [64]. Lameri et al. proposed an idea that CNN can be trained on synthetic data generated by gprMax [65]. In addition, they found that adding some background GPR acquisitions to the dataset can improve the rate of detection.

In addition, a CNN-based system can perform better with traditional preprocessing methods. Dinh et al. proposed a system combining CNN with conventional image processing methods, including migration and thresholding; this system could reach more than 95% in detection accuracy of rebars in bridge decks [66].

Another challenge in the application of CNN in GPR data is that GPR images are affected by buried soil. The same burial object can show different characteristics in different soils. E. Aydin et al. used multi-task learning (MTL) to solve this problem in 2019. This study combines the two single tasks of object detection and soil type detection. In addition, the soil type detection is used to assist the target detection. The MTL method increases the classification rate effectively [67]. The dependence on manual selection of parameters for GPR image preprocessing is another obstacle to automatic detection. Selecting an appropriate plot scale for the GPR image can enhance the target. Even in the same GPR survey data, the plot scale of different road segments is manually set to different values to generate suitable feature-enhanced GPR images. In this regard, J. Zhang et al. proposed a random incremental sampling (IRS) method referring to the particle filter method. This method generates a batch of random plot scale images for each segment of GPR data and inputs them into the deep learning model. If there are enough random images, the average value can be considered as the true value, so as to realize the automation of GPR system [68].

The R-CNN series algorithm is a classic two-part object detection algorithm. Recently, Faster-RCNN was applied to the detection of targets based on B-scan data in several

studies for its high efficiency. In 2018, Pham et al. transferred the weights of a CNN pretrained on grayscale Cifar-10 into a Faster-RCNN framework and then trained it with real and simulated GPR data. It is proven in this paper that Faster-RCNN outperformed traditional algorithms such as HOG and Haar-like [69] in feature extraction. In 2020, a study aimed at pavement distress detection discussed the performance of cascade CNN, region CNN, and Faster-RCNN, and the result indicated the possibility for a real-time detection with Faster-RCNN [70]. Some other researchers used some auxiliary methods to improve the performance of Faster-RCNN. A work employed various strategies, such as feature cascade, adversarial spatial dropout network (ASDN), Soft-NMS, and data augmentation, to improve the recognition accuracy of Faster-RCNN model for railway subgrade defect [71]. Another study in 2019 also adopted a data augmentation strategy in automatic hyperbola detection based on Faster-RCNN [72]. In 2022, H. B. Hu et al. used the Faster R-CNN model to identify and locate underground pipelines and used the Attention-guided Context Feature Pyramid Network (ACFPN) to optimize the feature extraction and used the cascade structure to improve the detection frame regression accuracy. The average recognition accuracy of the proposed model can reach 0.9256. This paper compares the proposed model with Faster R-CNN, YOLO-v5, and Mobilenet-SSD to demonstrate the superiority of the proposed model [73]. In 2022, Cui, F. et al. applied Faster R-CNN to the detection of underground media layering in highways. The model can effectively detect the subsurface interface of the highway, and the accuracy of the model can reach 98.30% [74]. In 2021, F. F. Hou et al. used Mask R-CNN to detect and segment semantic features. This paper introduces distance-guided intersection over union (DGIoU) as a new loss function for detecting and segmenting semantic features, overcoming the shortcoming of intersection over union (IoU) in training and evaluation. This paper also proposes a new method for extracting data points, which is robust to the interference of adjacent object features and background noise [75]. The network structure of [75] is shown in Figure 13. In the same year, F. F. Hou et al. used mask scoring R-CNN (MS R-CNN) as the main framework to monitor and segment target features. They used custom aspect ratio anchor boxes to improve network detection performance and used transfer learning method to improve the problem of insufficient training data. The feasibility of the method is verified in the actual data monitoring [76]. In 2023, Liu, Z et al. used Mask R-CNN to detect vertical cracks in asphalt pavement and measure the crack width. The average error of the width measurement is only 2.33% [77].

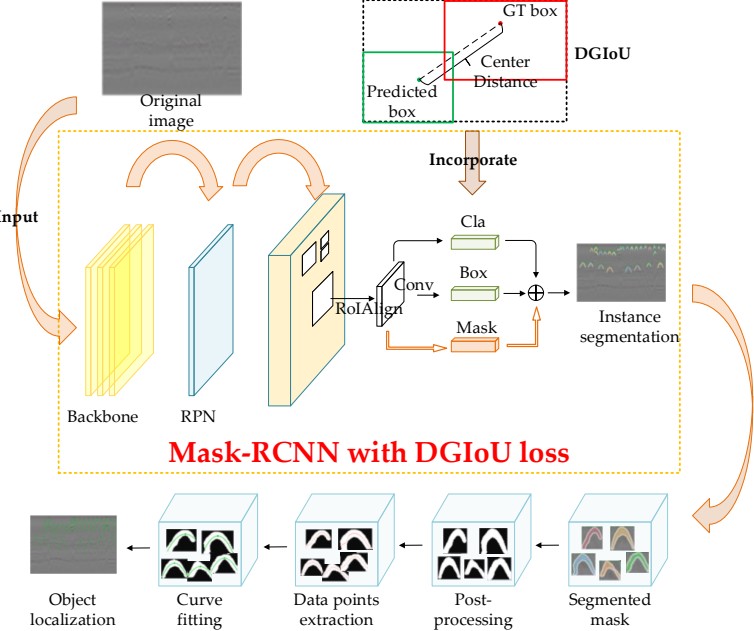

**Figure 13.** The network structure in [75].

There were other deep learning algorithms applied to GPR detection and which achieved good performance. In 2014, a study utilized a deep belief network (DBN), which can extract large, noisy, and complex datasets into compact feature sets, for detecting BEHs and achieved 91% probability of detection with 1.4 false alarms per square meter [78]. In 2022, DBN was also used to study factors affecting ground-penetrating radar signals when detecting reinforcement in concrete structures [79]. Another landmine detection adopted a convolutional autoencoder to find anomalous signatures [80]. In addition, without data of landmine reflection in the training set, this method could avoid missing novel landmine models. SSD is another important deep learning model that can produce fixed-size bounding boxes and score for the possibility of the target's presence. It was used in the detection and localization of rebar in concrete and proven to be seven times faster than Faster-RCNN with comparable ROC curves [81]. The network structure of [81] is shown in Figure 14. If we improve on the basis of SSD. Adding the feature pyramid network (FPN) to the whole network and using generalized intersection over union (GIoU) loss as the loss function can further improve the performance of the network [82]. Long short-term memory (LSTM) network has also been applied to GPR signal processing. In 2020, CNN-LSTM was also applied to the detection of underground cylindrical objects and outperformed single CNN or single LSTM [83]. In 2021, U. Ozkaya applied the residual CNN + Bi-LSTM model to the analysis of GPR B-scan images. This model is capable of automatically predicting GPR type, soil type, and scan frequency with high accuracy and fewer parameters, which has the potential for improvement [84]. The network structure of [84] is shown in Figure 15. In the same year, Juan, H et al. used LSTM to detect radar wavelets in the region that were used in traditional clutter-filtering algorithms [85]. In 2022, J. Wang et al. also combined the convolutional neural network based on DenseUnet and the recurrent neural network based on bidirectional convolutional long short-term memory to establish a new network model. The network performs both inversion and object recognition. Since the features of these two tasks are highly shared, the computational complexity can be greatly reduced. This paper validates the model at two levels. Synthetic data based on defect models are used first, and then sandbox models are exploited for testing in reality. The proposed model achieves good results on both datasets [86]. In recent years, generative adversarial networks (GAN) have also been gradually applied to GPR image detection. In 2021, Y. P. Yue et al. used an improved least squares generative adversarial network to generate GPR images to solve the problem of insufficient data. Compared with DCGAN and LSGAN, the images generated by the improved LSGAN are clear, and can learn hyperbolic characteristics better and show more details [87]. In 2022, Z. K. Ni et al. used conditional generative adversarial networks (cGANs) to complete clutter suppression of GPR images. In this method, the generation network processes the cluttered image into a clutter-free image, while the detection network takes the paired cluttered and clutter-free data as input and determines whether the clutter-free data are from the generation network. Since the network needs pairs of clutter-free images and clutter-free images as input and the real clutter-free image is difficult to obtain, the clutter image is obtained by superadding the simulated target image and the real clutter. The trained model produces good results on both simulated and real datasets. Compared with the traditional clutter removal methods, this method has advantages in computational complexity, clutter suppression effect, and applicability [88]. In the same year, Hu, D et al. innovatively adopted generative adversarial networks to enhance the synthesized GPR data and used U-Net inversion to obtain the subsurface permittivity map. In this study, GAN greatly improves the realism of synthetic radar maps, extending the proposed method to reality [89]. In 2023, Yurt, R et al. developed a new method for detecting underground cylindrical burial objects using A-scan images and time–frequency regression model (TFRM). In this method, the short-time Fourier transform (STFT) of one-dimensional signals is used to provide the joint distribution of time and frequency information. In this study, the data collected by CST modeling and the data measured by building the actual model were used. This method has a significant advantage in computational efficiency [90].

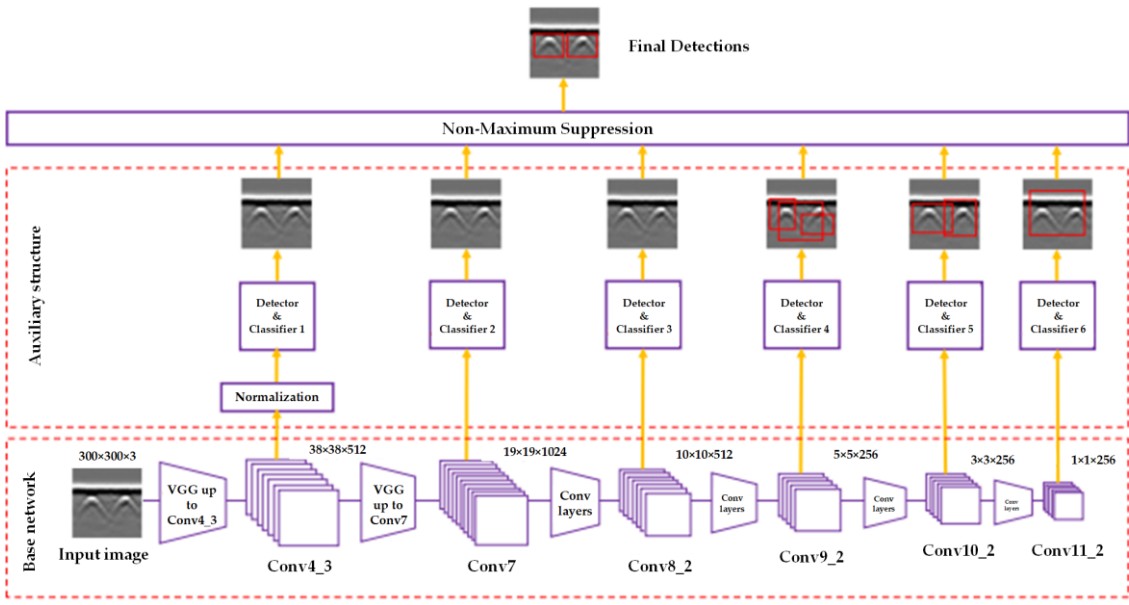

**Figure 14.** The network structure in [81].

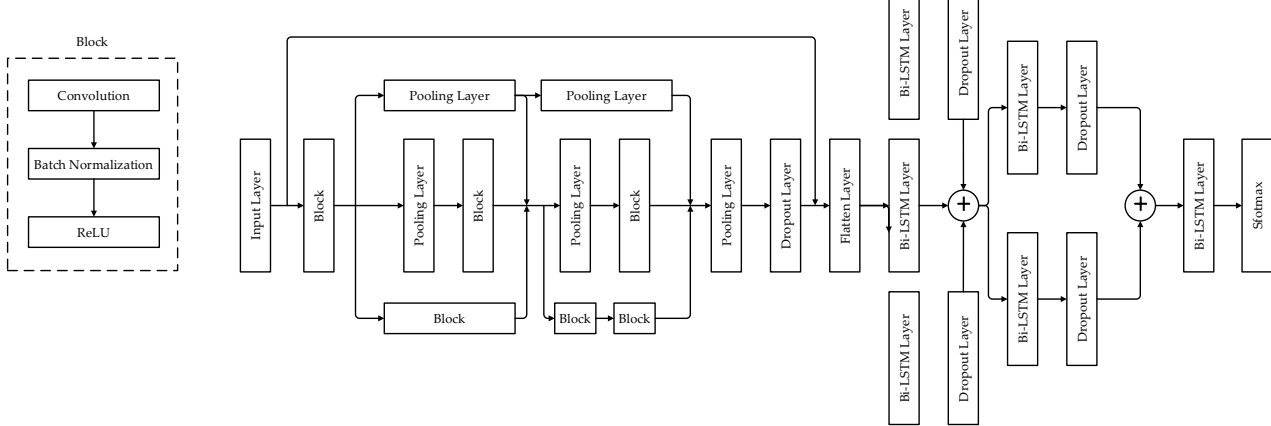

**Figure 15.** The network structure in [84].

As the technique of GPR is developing, the deep learning framework for detection based on C-scan data has been a direction of future development. In 2019, Kang et al. proposed an underground cavity detection network (UcNet) trained on 2D GPR grid images reconstructed from 3D GPR data [91]. A work from the same year designed a set of triplanar images composed of 2D images from three sections of raw GPR data as input of an AlexNet model, and the 2D images from three sections were put into RGB layers, respectively [92]. These two methods still used a 2D CNN structure while it could extract 3D features of images. Then, Khudoyarov et al. developed a real 3D CNN for underground object detection, and the average classification accuracy reached 97% in the detection of underground objects with 3D GPR data collected from urban roads in Seoul [93]. Another study applied a CNN-LSTM method in the BTD problems [94]. This study compared three different structures of CNN-LSTM based on GPR B-scan image in the down-track (DT) direction, B-scan image in the cross-track (CT) direction, and a fusion of both, respectively. Moreover, this work also discussed the performance of 3D CNN architecture. The results showed that CNN-LSTM outperformed the standard CNN in terms of ROC curve. In 2021, Q. Dai et al. used 3D U-net to establish the mapping relationship between C-scan images and 3D permittivity. Using the data obtained by simulation to verify the network, the mean absolute percentage error and structural similarity are 0.0994%

and 0.9980, respectively. In addition, the computational time required for this scheme to estimate the C-scan permittivity map is only 2.9 ms, which is at least 800,000 times faster than the classical reconstruction scheme [95]. In 2022, in the detection of underground defects of airport runway, N. S. Li et al. fused the 3D feature map of C-scan with the 2D feature map of top-scan. The multi-view features are combined by region alignment to complete the prediction target classification and regression. The paper is tested using a real-world runway dataset from three international airports. The test results show that the proposed method is superior to the state-of-the-art (SOTA) method and achieves a high detection rate in a variety of defect detections [96]. The network structure of [96] is shown in Figure 16. In the same year, M. Zhong et al. proposed a hybrid 3D electromagnetic full-wave inversion method for the reconstruction of underground detection targets. Compared with the pure FWI method BIM, the proposed method has higher accuracy and lower computational cost, while being more robust to noise for the reconstruction of multiple underground targets [97]. In 2023, Liu, Z et al. used YOLOv3 network with four-scale detection layer (FDL) to detect the combined image of B-scan and C-scan, from which road cracks were identified. Multi-scale fusion structure, efficient communicative loss function, K-means++ clustering, and hyperparameter optimization are used to further improve the detection performance. This research solves the problem of missed detection to some extent [98]. The network structure of [98] is shown in Figure 17.

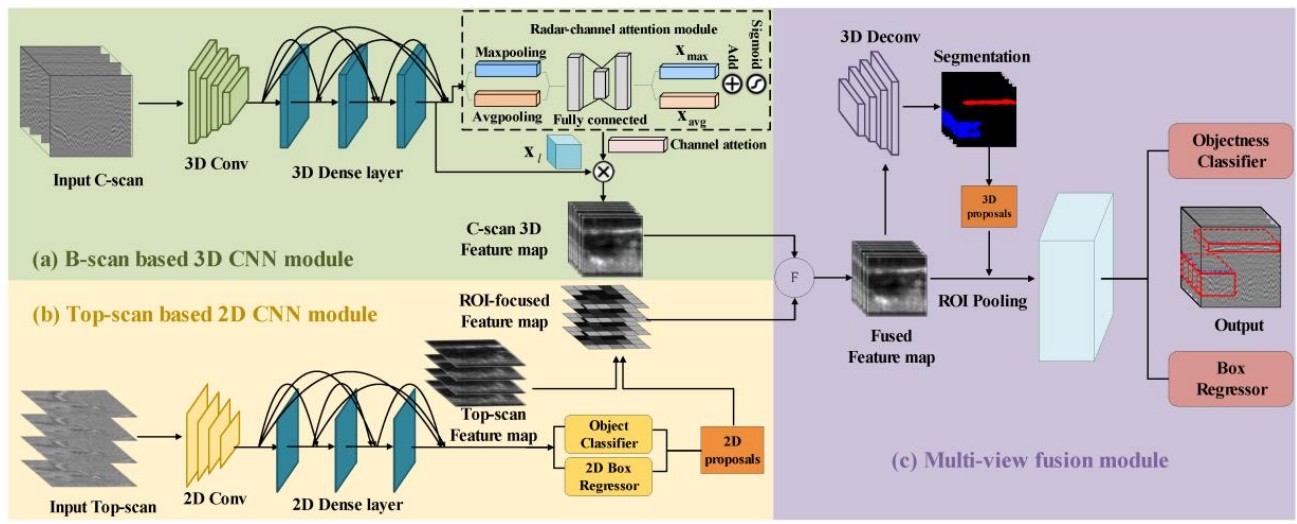

**Figure 16.** The network structure in [96].

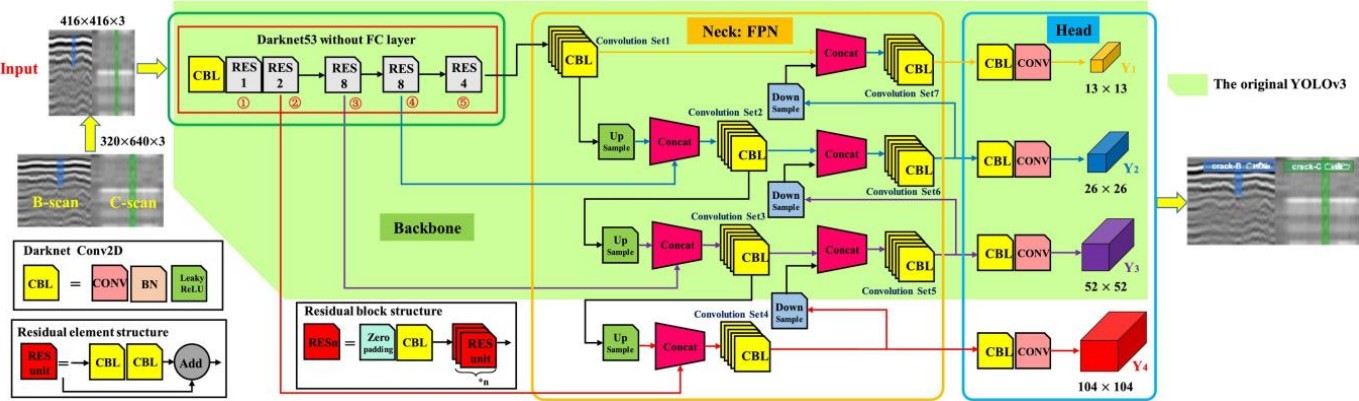

**Figure 17.** The network structure in [98].

In addition, Table 2 summarizes the works using CNN and other deep learning methods, listing their data sources and data form.

**Table 2.** The application of CNN and other deep learning methods in GPR detection and classification.

| Model | Year | Data Source | Data Form | Improvement |
|---|---|---|---|---|
| CNN | 2015 [58] | Government Eastern Test Range | B-scan | cross-validation, network weight regularization, and "dropout" |
| CNN | 2015 [59] | real data from a sensor array mounted on the front of a moving vehicle | B-scan | data augmentation |
| CNN | 2016 [60] | Eastern Test Range and Western Test Range | B-scan | cross-validation, weight regularization, dropout, pretraining, and data augmentation |
| CNN | 2018 [61] | simulated data | B-scan | |
| CSVM | 2020 [62] | simulated data and real data from GPR | B-scan | Combining CNN and SVM |
| CNN | 2017 [63] | real data from 8 distinct lanes at a U.S. test site using a vehicle mounted GPR system | B-scan | pretraining |
| CNN | 2017 [64] | real data from regular intervals on the path of the vehicle | B-scan | pretraining and data augmentation |
| CNN | 2017 [65] | synthetic data generated by gprMax and real-data from GPR acquisitions | B-scan | |
| CNN | 2018 [66] | real data from 26 bridge decks | B-scan | migration and thresholding |
| CNN | 2019 [67] | simulation data | B-scan | multi-task learning and transfer learning |
| CNN | 2020 [68] | real data from the Ningbo beltway | B-scan | incremental random sampling |
| CNN | 2019 [99] | real data from a newly renovated building | B-scan | |
| CNN | 2017 [100] | real data from freeway tunnel in Guangxi | A-scan | Wigner distribution |
| CNN | 2019 [91] | real data from urban roads in Seoul, South Korea | C-scan | UcNet |
| CNN | 2019 [92] | real data from urban road pavements in South Korea | C-scan | using triplanar images composed of 2D images from three sections of raw GPR data |
| CNN | 2020 [93] | urban roads in Seoul, South Korea | C-scan | 3D CNN |
| CNN | 2021 [95] | simulation data | C-scan | 3D U-Net |
| CNN | 2022 [96] | simulated data, artificial runway data, and real airport runway data | C-scan+top-scan | 3D CNN, Multi-view fusion |
| CNN | 2022 [97] | simulation data | C-scan | 3D U-Net |

**Table 2.** *Cont.*

| Model | Year | Data Source | Data Form | Improvement |
|---|---|---|---|---|
| CNN | 2022 [98] | real data from a 5.0 km section of a highway (G210) in Zhejiang Province, China | C-scan+B-scan | YOLOv3-FDL, EIoU loss function and K-Means++ clustering |
| Faster-RCNN | 2018 [69] | real data from several sites in France and synthetic data generated by gprMax | B-scan | pretraining |
| Faster-RCNN | 2020 [70] | real data from four highways in Shanxi Province | B-scan | |
| Faster-RCNN | 2018 [71] | real data collected by the subgrade status inspection vehicle | B-scan | feature cascade, adversarial spatial dropout network (ASDN), Soft-NMS, and data augmentation |
| Faster-RCNN | 2019 [72] | synthetic data and on-site data | B-scan | data augmentation |
| Faster-RCNN | 2022 [73] | real data from Zhengzhou | B-scan | Attention-guided Context Feature Pyramid Network |
| Faster-RCNN | 2022 [74] | real data is measured in the field | B-scan | |
| Mask-RCNN | 2020 [75] | real data from a concrete bridge deck | B-scan | a new loss function based on distance guided intersection over union. |
| MS R-CNN | 2021 [76] | real data from UT Gardens, Knoxville, USA | B-scan | customize the anchor's aspect ratio |
| Mask R-CNN | 2023 [77] | simulation data and real data obtained from S11 provincial highway measurements in Jinhua City, Zhejiang Province | B-scan | improved FPN structures |
| DBN | 2014 [78] | Eastern Test Range (ETR) and Western Test Range (WTR) | B-scan | |
| DBN | 2022 [79] | the measured data of the built model | B-scan | |
| convolutional autoencoder | 2018 [80] | real data from two different test sites | B-scan | |
| SSD | 2020 [81] | real data from residential buildings in two construction sites | B-scan | |
| SSD | 2021 [82] | simulated GPR data | B-scan | FPN, GIoU |
| CNN-LSTM | 2020 [83] | simulated GPR data and field GPR data from test site | B-scan | |
| CNN + Bi-LSTM | 2021 [84] | Real data | B-scan | |
| LSTM | 2021 [85] | simulated GPR data | B-scan | |
| CNN+ Bi-ConvLSTM | 2022 [86] | simulation data and sandbox model | B-scan | |

**Table 2.** *Cont.*

| Model | Year | Data Source | Data Form | Improvement |
|---|---|---|---|---|
| GAN | 2021 [87] | simulation data | B-scan | GPR image generation using GAN |
| cGAN | 2022 [88] | simulation data and real data | B-scan | the clutter is removed by means of cGAN |
| CNN-LSTM | 2020 [94] | real data from GPR mounted on the front of a moving vehicle | C-scan | |
| CNN+GAN | 2022 [89] | simulation data | B-scan | using GAN to improve image authenticity |
| TFRM | 2023 [90] | the data collected by CST modeling and the data measured by building the actual model | A-scan | |

## 5. Future Expectations

The future challenge and direction of development of GPR detection might be the following perspectives:

- Deeper, larger-scale neural network models. With the continuous development of deep learning technology and the continuous upgrading of hardware equipment, GPR image recognition will use deeper and larger scale network models to improve performance.
- Classification of different subterranean targets. For example, how to distinguish underground voids from infrastructures such as sewers without a pre-data acquisition process in terms of these infrastructures.
- Dataset preparation. CNN and other deep learning methods usually need a large volume of data for training, and the lack of training dataset can lead to an over-training issue. The proper application of synthetic data generalization and data augmentation for GPR can be a direction of study.
- Adaptive performance. Factors such as soil type and soil layer thickness can affect the GPR echo image. How to make the system adapt to different working environments and working scenarios is also a problem worth studying.
- Real time. With the development of technology, GPR should achieve real-time performance improvement to meet the needs of real-time detection.
- Detection based on C-scan data. Though there were several studies discussing the feature extraction of C-scan data and achieving some promising progress, more effective methods remain to be discovered.

## 6. Conclusions

GPR is a nondestructive tool for the investigation of the shallow subsurface, with a promising future in the detection of underground targets. Conventional ML, CNN, and some other deep learning methods have been widely used in the detection and classification of underground targets. Most of the conventional ML algorithms consist of feature extraction and classifier, while deep learning algorithms are usually end-to-end methods. SVM and K-NN are two popular classifiers in GPR detection for their high level of adaption and excellent performance. In addition, HOG, EHD, SIFT, and Gabor filter are widely used feature extraction techniques in these fields. DL is another flexible method to represent the features of the dictionary that can be gained by learning. CNN has attracted more attention since the deep learning framework based on CNN was proposed and achieved remarkable performance. CNN is one of the most promising methods in the detection and recognition of underground targets, as it can function well without artificial features. As

an increasing number of deep learning algorithms are being proposed, other techniques, such as SSD, LSTM, DBN, and GAN, have also been employed for the recognition and detection of GPR data with notable success. Concurrently, in order to improve the accuracy of target recognition and detection, the form of GPR data processed by the algorithm has gradually changed from B-scan to C-scan, which contains three-dimensional information of underground structures. Recognition and detection based on C-scan images will also become an important research direction in the future.

**Author Contributions:** Methodology, X.B. and S.W.; formal analysis, G.C. and H.L.; investigation, Y.L. and H.T.; writing-original draft preparation, Y.Y., T.Z. and H.C.; writing-review and editing, Y.Y. All authors have read and agreed to the published version of the manuscript.

**Funding:** This work was supported by National Natural Science Foundation of China under Grant 62071147.

**Institutional Review Board Statement:** Not applicable.

**Informed Consent Statement:** Not applicable.

**Data Availability Statement:** The data used in this article is not public.

**Conflicts of Interest:** The authors declare no conflict of interest.

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
