# Peer review of "A Comprehensive Review of Conventional and Deep Learning Approaches for Ground-Penetrating Radar Detection of Raw Data"

_applsci, doi:10.3390/app13137992_

Round 1
Reviewer 1 Report
A Comprehensive Review of Conventional and Deep Learning Approaches for Ground-Penetrating Radar Detection:
Comments:
The manuscript is added value to the GPR geophysical technique, especially by application of the new trends of Deep Learning Approaches
Figure 10: need some clarification, and addressing the components of the figure.
Figure 11: need some clarification, and addressing the components of the figure.
Figure 12: it is favorable to insert the reference name (not the reference number.
Figure 13: it is favorable to insert the reference name (not the reference number.
Figure 14: it is favorable to insert the reference name (not the reference number.
Figures 15. 16, 17: it is favorable to insert the reference name (not the reference number.

Reviewer 2 Report
This article presents a literature review of Conventional and Deep Learning Approaches for Ground-Penetrating Radar Detection, as well as a summary of machine learning and reinforcement learning algorithms applied in the field, the contribution of the paper is clear and of general interest. Here are some comments:
· the text has some spelling errors and lack of punctuation, as in line 31
· in section 1.2, it is recommended to add references that support the theory that shows.
· In section 3.2, authors are recommended to add updated references for the year 2022 and 2023, to express the most recent advances in the field.
· The authors must explain the methodology used to select the ML and deep learning algorithms used in the review, what were their delimitations? Did you use any methodology as a prism?
· In general, the text is well written, it provides relevant information about the field, however, it is recommended to reinforce the methodological part of the research.
·
Minor editing of English language required
Reviewer 3 Report
The paper deals with a review of the application of ML and DL algorithms to the processing of radar data in GPR applications. SInce it deals only with radargrams under different scan modalities, I believe that the title is a bit misleading and a more appropriate one would include (for instance at the end) the words either "from radargrams" or "form raw data". Also the key words should be expanded to include one of these words.
As a general comment, I note that no mention is made of other GPR detection techniques such as the tomographic ones, based on more accurate electromagnetic modelling of the wave interactions, or the ones based on signal subspace projection methods, as the MUSIC algorithm. Therefore the authors should correctly set their review work in the more general framework of GPR processing.
Moreover, references needs attention. At first page: [1] does not correspond to nothing, while [3] seems unappropriate.
At the same time ,at page 26, I do not understand [1] what corresponds to , and [3] seems uncorrectly quoted as it deals with a review.
In ref [5], what the double X refer to?
In general, the format of the references must be standardized: sometimes the journal names are shortened; they are quoted in capital, italic or standard type without a rule; the same holds for the proceedings.
Round 2
Reviewer 2 Report
the authors complied with the recommendations given
Reviewer 3 Report
The authors have accepted almost all my suggestions. However I believe that a sentence as "Tomographic techniques, based on more accurate electromagnetic modelling of the wave interactions, are missing from the manuscript", as the authors admit in Response 2, must be included in the introduction.
